# Computational Simulation of Adapter Length-Dependent LASSO Probe Capture Efficiency

**DOI:** 10.3390/biom9050199

**Published:** 2019-05-22

**Authors:** Jingqian Liu, Syukri Shukor, Shuxiang Li, Alfred Tamayo, Lorenzo Tosi, Benjamin Larman, Vikas Nanda, Wilma K. Olson, Biju Parekkadan

**Affiliations:** 1Department of Biomedical Engineering, Rutgers University, Piscataway, NJ 08854, USA; jl2085@scarletmail.rutgers.edu (J.L.); lorentosi@gmail.com (L.T.); 2Department of Surgery, Center for Surgery, Innovation and Bioengineering, Massachusetts General Hospital, Harvard Medical School and the Shriners Hospitals for Children, Boston, MA 02114, USA; sykrishukor@gmail.com (S.S.); alfredgtamayo@gmail.com (A.T.); 3Department of Chemistry and Chemical Biology, Rutgers University, Piscataway, NJ 08854, USA; shuxiang.li@rutgers.edu (S.L.); wolson@connect.rutgers.edu (W.K.O.); 4Division of Immunology, Department of Pathology, Johns Hopkins University, Baltimore, MD 21205, USA; hlarman1@jhmi.edu; 5Center for Advanced Biotechnology and Medicine, Robert Wood Johnson Medical School, Rutgers University, Piscataway, NJ 08854, USA; nanda@cabm.rutgers.edu; 6Harvard Stem Cell Institute, Harvard University, Cambridge, MA 02138, USA

**Keywords:** gene-capture probe, LASSO probe, oxDNA, VMMC simulation, DNA–DNA interaction

## Abstract

Multiplexed cloning of long DNA sequences is a valuable technique in many biotechnology applications, such as long-read genome sequencing and the creation of open reading frame (ORF) libraries. Long-adapter single-stranded oligonucleotide (LASSO) probes have shown promise as a tool to clone long DNA fragments. LASSO probes are molecular inversion probes (MIP) engineered with an adapter region of user-defined length, flanked between template-specific probe sequences. Herein, we demonstrate that the adapter length is a key feature of LASSO that influences the efficiency of gene capture and cloning. Furthermore, we applied a model based on Monte Carlo molecular simulation in order to study the relationship between the long-adapter length of LASSO and capture enrichment. Our results suggest that the adapter length is a factor that contributes to the free energy of target–probe interaction, thereby determining the efficiency of capture. The results indicate that LASSOs with extremely long adapters cannot capture the targets well. They also suggest that targets of different lengths may prefer adapters of different lengths.

## 1. Introduction

There is a dramatic asymmetry between genomic sequence information and the functional interpretation of that data, in part due to a lack of cost-efficient techniques for synthesizing stretches of DNA that are longer than 400 base pairs (bp). Long-adapter single-stranded oligonucleotide (LASSO) probes were developed as a promising tool to fill this technological gap by cloning long genomic DNA targets in a parallel and multiplexed manner [1]. The mature probe is comprised of a linear single oligonucleotide consisting of an adapter sequence between two probe arms that are complementary to the two ends of a target DNA fragment, a design that is similar to traditional molecular inversion probes (MIP) used for single nucleotide polymorphism analysis, except that the adapter length (AL) can be controlled [1]. The adapter sequence enables a LASSO probe to form a padlock shape upon specifically hybridizing the two arms to a target genomic region of interest [2,3]. The two arms are named the extension arm where the polymerase binds and the ligation arm where the gap is filled [1]. LASSO probes have been shown to be far more effective than traditional MIPs at capturing DNA targets longer than 400 and up to 5000 bp [1,4,5]. The adapter regions of LASSO probes, which have been empirically optimized for given target sizes, warrant further investigation.

Previous experiments demonstrated that LASSOs with ALs of ~200 and ~400 nucleotides (nt) provided higher gene-capture efficiency compared to MIPs with ALs of ~50 nt [1]. Inspired by these results, we offered the hypothesis that longer ALs should be capable of capturing longer DNA targets. In order to test this hypothesis, we set out to study the relationship between AL and target capture efficiency by combining computational simulations with classical molecular biology techniques. In this study, four ALs of ~50 nt (typical adapter length of MIPs), ~200 nt, ~400 nt, and ~800 nt were examined experimentally. Probes with different ALs were applied to capture single target or multiple targets in a gene library, where different capture efficiencies could be observed. We adopted a coarse-grained model, oxDNA [6], to simulate a large time and spatial scale of DNA–DNA interactions. We used Virtual Movement Monte Carlo (VMMC) simulation [7,8] to estimate the free energy barriers for target–probe interaction and compared the results with the relative free energy deduced from experimental data. Herein, we report predictions of the efficiencies of LASSO capture based on the simulations. We show that the probe–target interaction free energy depends upon AL and captures some of the experimental findings. This in silico model provides a foundation for improved computational design of LASSOs for new gene target libraries.

## 2. Materials and Methods

### 2.1. Generation of Long-Adapter Single-Stranded Oligonucleotide Probe & Target Cloning

In single-target cloning experiments, LASSO probes were designed with permutations of 242-, 442-, and 788-nt long adapter sequences flanked by probe arms targeting 0.6, 1.0, and 2.0 kilobase (kb) regions within the single-stranded M13mp18 bacteriophage genome (NEB). The exact lengths of the adapters depend on primer-binding sites on the template plasmid pCDH-CMV-MCS-EF1-Puro (SBI System Biosciences, 2438 Embarcadero Way, Palo Alto, CA, USA), which is used to produce long adapters. Omitting de novo LASSO probe synthesis to minimize improperly formed LASSO probes that may add to non-specific PCR products, a total of nine LASSO probes were synthesized as double-stranded gBlocks (IDTDNA). These probes were then PCR amplified with inversion PCR primers TioINewRoche and SapINew (5-Phos) (IDT, Integrated DNA Technologies, Coralville, IA, USA). Single-target captures were done with ten-fold molar excess of probes to genomic DNA to optimize capture parameters.

*Escherichia coli* ORFeome LASSO probe library assembly and captures were done using a previously established protocol suited for massively multiplexed purposes [1]. Post-capture PCR amplification was the same for both *E. coli* ORFeome and single target captures. All sequences used in both ORFeome library experiments and single-target capture validations are included in Appendix A (Appendix A).

### 2.2. LASSO Capture Quantification for Single-Target and Library Experiments

LASSO capture PCR products for single-target experiments were visualized via polyacrylamide gel electrophoresis (PAGE) (Thermo Fisher Scientific, Waltham, MA USA) stained with 1X SYBR Gold nucleic acid stain. Obtained gel images were then processed with ImageJ (Fiji) [9] to quantify AL-dependent LASSO capture efficiency. PCR product mass measurements obtained within the linear range of standard DNA ladders were averaged and then tabulated using Microsoft Excel v.16.16.3 (181015) (Redmond, WA, USA) (see in Appendix A).

The quantification of LASSO efficiency in library experiments was based on DNA sequencing data of target–cloning products. “Enrichment” of the probes was evaluated as the ratio of the median RPKM (reads per kilobase per million mapped reads) [10] of target sequences and the median RPKM of off-target sequences. RPKM of target sequences can be calculated by Equation (1), where $reads_{target}$ represents the reads value corresponding to the target, $reads_{lib}$ represents the total reads value of the whole library, and $l_{target}$ represents the number of bases, or length, of the target.
(1)$$RPKM=\frac{reads_{target}}{reads_{lib}\cdot l_{target}}.$$

### 2.3. Probe–Target Interaction Simulation

#### 2.3.1. Order Parameter

Two distances involving complementary bases formed by the extension arm and the ligation arm were used to describe the state of the probe and target gene. The respective distances, $d_{ex}$ and $d_{lig}$ (see Figure 1A) can be further simplified to two order parameters $Q_{ex}$ and $Q_{lig}$. The order parameters, which range from 0 to 10, represent the values of the distances within the following 11 intervals: 0–1.7, 1.7–3.4, 3.4–5.1, 5.1–8.5, 8.5–12.8, 12.8–17.0, 17.0–34.1, 34.1–68.1, 68.1–102.2, >102.2 nm. For instance, a $Q_{ex}$ value of 0 indicates that $d_{ex}$ is smaller than 1.7 nm, corresponding to a probe–target interaction on the site of the extension arm. By contrast, a $Q_{ex}$ value of 10 depicts a state in which $d_{ex}$ is larger than 102.2 nm, indicating that no probe–target interaction occurs on the extension site according to our definition. Figure 1B shows examples of non-interaction and interaction states. When both $Q_{ex}$ and $Q_{lig}$ are 10, the two strands are in a completely non-interaction state. The sampled configuration is considered an interaction state when both $Q_{ex}$ and $Q_{lig}$ values equal zero (see Figure 1B).

#### 2.3.2. Umbrella Sampling

Umbrella sampling [7,11,12] was applied to sample the states of probe–target interaction. Weights for different states could be manually set, to control the probability of state transition. For example, given weight of state 0 (corresponding to order parameter $\left(Q_{ex}{}_{0},\;Q_{lig}{}_{0}\right)$) as $w_{0}$ and the weight of state 1 (corresponding to order parameter $\left(Q_{ex}{}_{1},\;Q_{lig}{}_{1}\right)$) as $w_{1}$, the probability of a state transition from state 0 to state 1 is adapted by the factor $\frac{w_{1}}{w_{0}}$. Therefore, state 1 can be easily sampled even if the energy change from state 0 to state 1 is unacceptable. After sampling was finished, the diagram which stored the frequency of each state was transformed to an unbiased distribution (see Figure 1C) to eliminate the effect of the chosen weight. Finally, the free energies were calculated from the unbiased distribution data (see Figure 1D).

#### 2.3.3. Box Size and Salt Concentration

The box side dimension was set in accordance with the sizes of the targets and probe in order that both were well contained inside the box. To reflect the salt environment of 10 mM $\left[{\mathrm{Mg}}^{2+}\right]$ and 100 mM $\left[{\mathrm{Na}}^{+}\right]$, the sodium concentration was set as 300 mM (since there is no magnesium concentration included in oxDNA, we used a higher concentration of ${\mathrm{Na}}^{+}\;$as a substitute, according to the results of known reports which show similar features of DNA shared by the conditions of certain magnesium concentration and of much higher sodium concentration [13].).

#### 2.3.4. Procedure of Simulation

The whole simulation could be sequentially divided into three stages: First, the weight assigned in the biased sampling was adjusted iteratively by a shell script until configurations under each state could be uniformly sampled and could easily be transformed from one to another. This means the counts over each histogram bin should be comparable so that the whole histogram could be considered flat. Second, the aforementioned weights were applied for equilibria. Finally, sampling began to yield a free energy profile and further estimate the energy barrier that needs to be crossed for the capture of target genes.

## 3. Results

### 3.1. Adapter Length-Dependent LASSO Probe Efficiency

As previously reported, LASSOs hold superiority over traditional MIPs in cloning long genomic targets [1]. The hypothesis that the efficiency of pre-cloning capture is a function of AL was tested using single targets of known sizes while varying the AL (242 nt, 442 nt, and 788 nt) of a LASSO probe for the target. Experiments for three single targets of 0.6, 1.0, and 2.0 kb length were conducted to obtain a clearer observation of capture efficiency as a function of LASSO AL. Products of capture, namely the target–probe complexes, were subject to PAGE (Figure 2A). PAGE DNA quantification in Figure 2B demonstrated a two-tailed distribution of capturing efficiency for different ALs on each LASSO target. *p*-Values were respectively measured as 0.0068, 2.5 × 10^−7^, and 4.1 × 10^−6^ within the group of 0.6-, 1.0-, and 2.0-kb targets. Furthermore, the preference of AL also changes as target length varies. As the results show, targets 0.6 and 2.0kb in length prefer 442-nt adapters while 1.0 kb targets gain more captures with a 242-nt adapter. These results for single targets were also verified in library experiments involving the simultaneous capture of thousands of DNA targets with lengths varying from 400 to 5000 bp. LASSOs or MIPs with different adapters and the same extension and ligation arms were tested to capture different genes in the *E. coli* genome in independent experiments. As a pooled effect, probes with 442-nt AL provided the highest capture enrichment, found in the sequences produced by multiplex-DNA cloning. (Figure 2C) For all targets studied, MIP and LASSO with a 788-nt adapter provided less favorable efficiency compared with the other two LASSO adapter sizes. These empirical results suggest that longer LASSO probes are not necessarily better for a given target size.

### 3.2. The Effect of Secondary Structure Formed by LASSO

The efficiency of capture could be influenced by many factors. One hypothesis which may lead to the failure of the 788-nt adapter LASSO is the potential complex secondary structure that the probe might form due to its considerable length. Mfold [14] was applied to predict the secondary structures of the DNA LASSOs. Computations were conducted using input information representing the experimental LASSO conditions (*T* = 65 °C $\left[{\mathrm{Mg}}^{2+}\right]=10\;\mathrm{mM}$, $\left[{\mathrm{Na}}^{+}\right]=100\;\mathrm{mM}$) corresponding to the results shown in Figure 2. The Mfold outputs show extremely complex secondary structures at 37 °C (tested as a reference) but largely reduced motifs at the experimental temperature (Figure 3).

The contrast between structure prediction at 37 °C and under the experimental conditions of LASSO target capture suggests that the secondary structure, which may intervene in target–probe interaction at the lower temperature, is nearly eliminated by the high temperature of the experiment. Most of the predicted motifs only involve bases within a short range, thereby preventing dramatic deformation of the whole probe. Also, the percentage of secondary structures does not show obvious differences as the AL is changed. Through this observation, we conclude that secondary structure formation is not the main factor that underlies the discrepancy of efficiency.

### 3.3. Calculation of Free Energy of Probe–Target Interaction

As an alternative hypothesis, the mechanical properties of single-stranded DNA, such as the persistence length [15], can contribute to the final efficiency of target capture. In the scenario of capturing a DNA target, an improper probe length could lead to the mismatch of the end-to-end distances of the probe and target sequences. Moreover, conformational entropy may contribute to the performance of the probes. For instance, a longer probe may be ineffective due to the entropic cost of interacting with target sequences. We used computational simulations to understand this issue by estimating the free energy of probe–target interaction, which includes the effects of entropy.

To quantify the free energy of target–probe interaction, VMMC simulations were conducted with oxDNA [6], software especially designed for the study of the mechanical properties of DNA molecules. Its coarse-grained model enables the simulation of large systems under long time scales [16] and thus provides an appropriate tool for our study of two long strands of DNA. Furthermore, studies about DNA hybridization [17,18,19] and DNA cyclization [7,20] have been successfully conducted with oxDNA. In our simulations, we adapted reaction coordinates to fit the case of LASSO probes and followed the same procedures used in prior work with oxDNA.

The system was simplified to describe a local scenario. The target was treated as a DNA piece without the constraint from the whole genome since fragmented DNA is typically input into a LASSO capture reaction volume. A single-stranded structure was adopted, considering the high temperature of the experiments. The VMMC algorithm [7,8] was applied here to avoid the low acceptance of proposed movements of the two DNA strands and was used in conjunction with umbrella sampling [7,11,12] to access rare states of target–probe interaction.

The two-dimensional order parameters ($Q_{ex},Q_{lig}$), which represent the reaction coordinate, were used to describe the states of the two DNA strands. $Q_{ex}$ is the parameter representing the minimum distance between the bases on the extension arm of the LASSO and the complementary bases on the target site. $Q_{lig}$ is the corresponding parameter for the ligation arm. (Figure 1A) The values of $Q_{ex}\;$and $Q_{lig}$ range from 0 to 10, which respectively represent 11 intervals within which those distances may fall. The order parameters enable a description of the probe–target interaction state of the current configuration. For instance, $Q_{ex}=0$ demonstrates a state in which the extension arm of the probe LASSO and its complementary region on the target is closer than 1.7 nm. $Q_{ex}=10$ means the configuration is in a state where this distance falls in an interval larger than 102.2 nm. A state that involves a target–probe interaction is defined as a state that satisfies the requirement of $Q_{ex}=0\;$and $Q_{lig}=0$. By contrast, with both $Q_{ex}$ and $Q_{lig}$ value larger than 10, the configuration will be treated as a non-interaction state. The difference between the two states in oxDNA simulations is shown in Figure 1B. A close-up view is provided to display the interaction between the probe arm and its complement on the target.

In the simulation, different weights were assigned to different states, which correspond to different values of the two-dimensional order parameters. A large weight is assigned to a rare state, which is generally rejected due to an unacceptable energy change, to increase the possibility of acceptance. Without such intervention, it is not possible to sample states of close probe–target interaction. After sampling, an unbiased sampling frequency can be gathered from the collective data of biased configurational samples and the weights originally assigned to the different states. An example of final unbiased sampling is shown in Figure 1C, which presents a slope-like distribution. The free energy is calculated from the two “corner” values, namely the two sampling frequency values of non-interaction states and states of interaction. The sampling condition on the cross-section of the diagonal of the slope (marked by the red line in Figure 1C) is reflected in Figure 1D. The $\Delta G_{int}$ represents the calculated free energy of target–probe interaction, which was calculated from equation:(2)$$\frac{\Delta G_{int}}{RT}=-ln\left(\frac{P_{\left(Q_{ex}\;=\;0,\;\;Q_{lig}\;=\;0\right)}}{P_{\left(Q_{ex\;}=\;10,\;\;Q_{lig}\;=\;10\right)}}\right).$$

### 3.4. Comparison with Experiments

To evaluate whether the simulations account successfully for the experiments, the experimental data were represented in the form of free energy differences. The values of RPKM (see in Appendix A) were quantified using next-generation DNA sequencing after library cloning of *E.*
*coli* genes, aiming to represent the enrichment efficiency for certain probes [21]. It is reasonable to believe that these values could reflect the free energy difference, although they are not parameters which are rigorously related to the free energies. Therefore, RPKMs of the cloned copies for one target under the application of different probes were transformed into “free energy differences” by Equation (3). This alternative method was adopted so that a quantitative comparison could be conducted between simulation and experiment. During this process, we assumed that the experiment is equivalent to adding four probes simultaneously into a pool and that there is a competition among the four probes to interact with the same gene sequence. A zero free energy level, which corresponds to the largest RPKM value, is regarded as the most favorable selection of AL and also as a zero free energy reference for simulations.
(3)$$\Delta G_{probeA-probeB}=-ln\left(\frac{RPKM_{probeA}}{RPKM_{probeB}}\right).$$

Six groups of target gene sequences and probes were subjected to VMMC simulation. The target genes were randomly selected from the library, with lengths ranging from 400 to 1500 bp, to ensure that different target lengths were covered in the study. Three of the target genes were chosen to have the same length (599 bp) in order to see if the model is sequence-sensitive. The RPKM data were further transformed to free energy differences in order to compare simulations and experiments. Figure 4A–F shows these results. Pearson correlation coefficients [22] were calculated to evaluate the correlation between the experimental results and the simulation predictions. Three out of six groups show relatively strong correlations (>0.7). Five out of six (~90%) provide predictions of AL preference consistent with experiments. The lowest free energy suggests the most favorable option of probes. As the target length increases, a transition in optimal AL from 54-nt (the MIP AL) to 242-nt and 442-nt could be observed (compare Figure 4A vs. Figure 4E,F). The difference between the free energy of MIPs relative to zero free energy reference slightly increases (from 0 to a positive value) when the target length changes from 400 to 1500 bp. The increased difference might be a clue as to why LASSOs exhibit more favorable efficiency in capturing and cloning long targets from a computational aspect. The predicted difference, however, is not consistent with the considerable difference between MIPs and LASSOs observed in the experiments. Furthermore, all the simulations showed that the probes with 788-nt adapters demonstrated larger probe–target interaction free energies than the probes with 242- and 442-nt adapters. This result provides a potential explanation for the unsatisfactory performance of 788 nt-adapter LASSOs. Along with the experimental results, it suggests that an extremely long adapter will not contribute to improving the efficiency of a gene-capture probe and that the most suitable AL should be within a certain range.

## 4. Discussion

LASSO probes can enable long-read DNA sequencing and multiplexed cloning efforts if they are highly robust and efficient. This study evaluated the adapter region of a LASSO probe, both experimentally and computationally, to determine the effects of AL on capture efficiency. For a given target size, it was determined that there was a distribution of efficiency for LASSO probes where intermediate adapter sizes led to the best capture results. These empirical results motivated an in-silico effort to understand LASSO probe binding and help account for these results. A coarse-grained model (oxDNA) reproduced the discrepancies in efficiency between different LASSOs and MIPs and provided a proof-of-concept that computational simulations could be a useful quality control check to design LASSOs in the future for optimal results given a pre-specified length of DNA target. There remain several open areas to build upon this first-generation computational model that can improve predictions.

Although the simulation results captured some of the experimental results and provided a reasonable explanation for some phenomena, such as the failure of 788-nt adapters, the computations require further improvement. For instance, the free energy predictions do not provide a sufficient computational rationale for the inferior performance of MIPs. The considerable interaction free energy gaps between MIPs and LASSOs shown experimentally in the cases of several targets (Figure 4B,E,F), could not be well captured. The simplicity of the model should be firstly taken into consideration if we start to analyze the experiment–simulation difference from the aspect of simulation. The free energy of target–probe interaction is presumed to be the main factor that causes the efficiency discrepancy. The post-interaction hybridization of bases was ignored. This term is considered to have an equal contribution to the total target-capture free energies of all the probes since the MIPs and LASSOs for one gene share the same extension and ligation arms. Nevertheless, the internal stress which varies as the probe length changes could have an influence on the unzippering energy [23]. Considering this, the term may also influence the capturing performance of the probes due to the different constraints during hybridization and may further account for the inefficient performances of the MIPs.

Moreover, the target was treated as a DNA fragment that could move without constraints from surrounding sequences. In reality, the target is likely located in large genome fragments due to the imprecision of sonicating DNA before a capture experiment is conducted. This factor may have an impact on the behavior of the target due to constraints associated with the long chains of the genome fragments. This might also be a clue behind some of the differences between experiment and simulation. The imprecise way of obtaining experimental relative free energy might also contribute to these differences. Finally, the experimental errors can also be a result of artifacts from PCR amplification of a captured target library and sequencing itself.

Another factor that may take account of the difference between Figure 2C and the simulation results is the intervention of non-target capture. The simulations were unable to include non-specific captures. Therefore, the model did better in predicting the most favorable probe for a specific target, but it was not capable of considering a probe with the least side effects of non-target capture.

In summary, we report a computational model of LASSO binding that roughly predicts experimental capture results. It is envisioned that, with further improvements of this computational model, such knowledge can help customize adapter sequences for future large library preparations and obviate the need for empirical determination of an optimal LASSO adapter size. By having further understanding and control of LASSO adapter sizes, there can be a better assurance of the deployment of LASSO probes in a wide range of applications where the versatility of these reagents will be necessary. Ultimately, this simulation engine can be built into a front-end user interface for the public to create LASSOs on their own with customized binding arms and ALs/sequences to enable broad adoption of the technique by the research community.

## Figures and Tables

**Figure 1 biomolecules-09-00199-f001:**
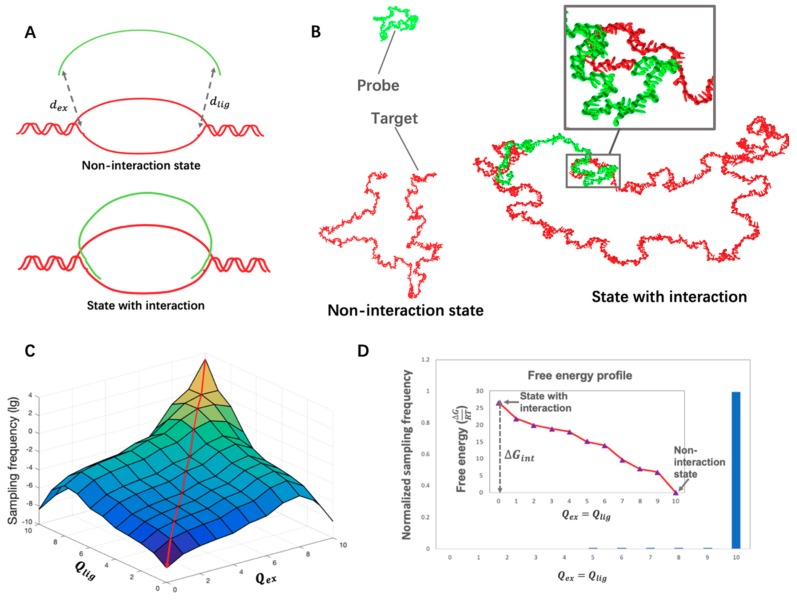
(**A**,**B**): Diagram and oxDNA visualization showing non-interaction/interaction state of the target and gene. (**A**) A diagram showing the non-interaction/interaction states. $d_{ex}$ and $d_{lig}$ respectively represent the distances between the extension arm and its complementary region on the target gene and the ligation arm and its complementary region on the target gene. These two parameters are respectively represented by $Q_{ex}$ and $Q_{lig}.$ (**B**) Non-interaction/interaction states of the target and the probe visualized by cogli1, a tool to draw oxDNA configurations. The figure on the left shows a complete open state of the target (colored in red) and the probe (colored in green) in which both $Q_{lig}$ and $Q_{ex}$ equal 10. The figure on the right demonstrates an interaction state of the target and probe where both $Q_{lig}$ and $Q_{ex}$ equal zero. Each visualized nucleotide includes two beads, one representing the backbone and the other the base. There are two interaction sites on the base, including a stacking site as well as a hydrogen-bonding site [11]. (**C**,**D**): Free energy calculation for the *E. coli* gene *nemR* and molecular inversion probe (MIP). (**C**) An unbiased sampling frequency distribution over the whole space where $Q_{ex}$ and $Q_{lig}$ range from 0 to 10. (**D**) The free energy profile along the diagonal plane of part **C** (marked by the red line). The value of the free energy reflects the change in distance along the diagonal of part **C**, where $Q_{ex}=Q_{lig}$. The biased sampling frequency and the calculated free energy profile are shown. $\Delta G_{int}$, the free energy of probe–target interaction is labeled in the figure.

**Figure 2 biomolecules-09-00199-f002:**
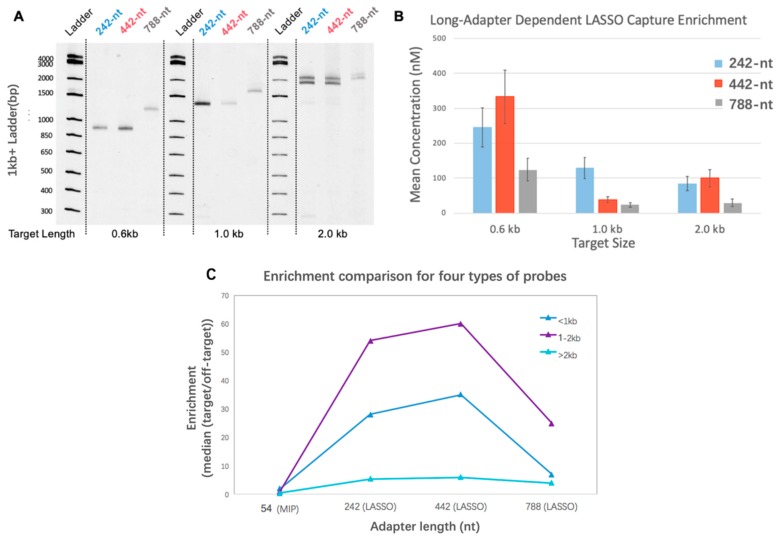
Dependence of capture efficiency on adapter length and target length. (**A**) Polyacrylamide gel electrophoresis (PAGE) visualization of amplified capture products for different adapter lengths and target sizes. (**B**) Single-target capture quantification from PAGE, each quantified with standard curves derived from DNA ladder serial dilutions with known band concentrations. P values that describe the confidence in the observed difference for different adapter lengths within each group of target sizes are measured to be 0.0068, 2.5 × 10^−7^, and 4.1 × 10^−6^. (**C**) Description of enrichment efficiency from sequences resulting from multiplex target cloning of *Escherichia coli* genes.

**Figure 3 biomolecules-09-00199-f003:**
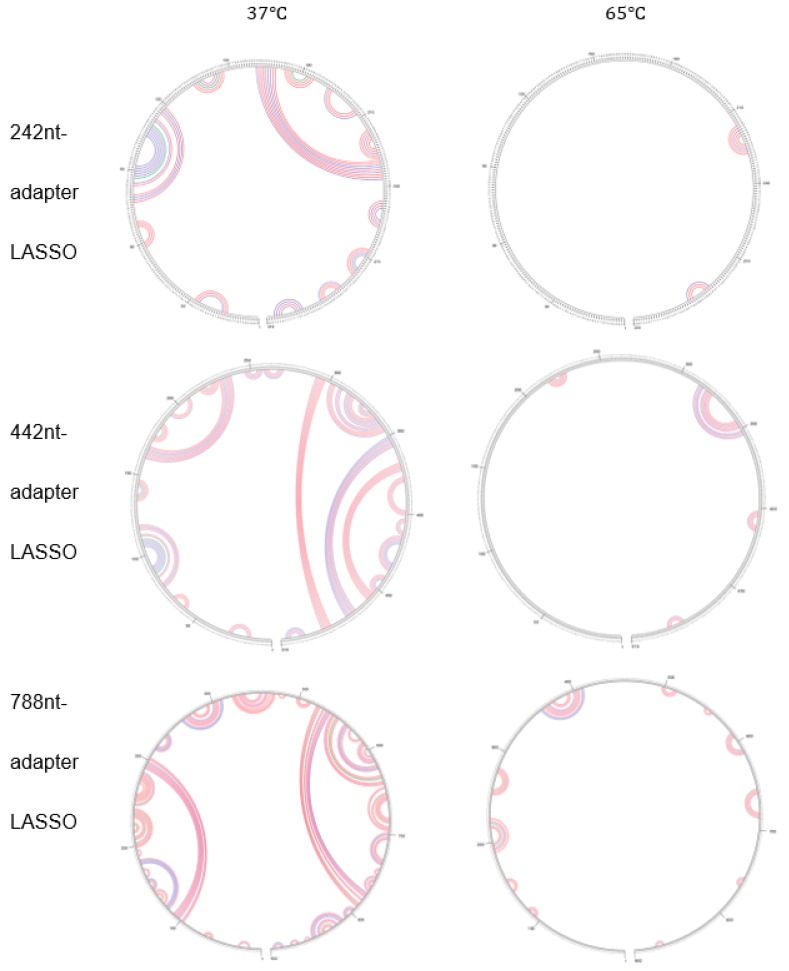
Circle graphs of secondary structure predicted by Mfold [14] for LASSO probes designed to capture the *E. coli* gene *nemR* at 37 °C and 65 °C. Bases are drawn along the circumference of the circle. The arcs inside the circles denote the base pairs predicted by Mfold to contribute to the secondary structure of the probe. The red arcs correspond to the predicted G—C pairs, the blue arcs to the A—T pairs and the green arcs to the G—T pairs.

**Figure 4 biomolecules-09-00199-f004:**
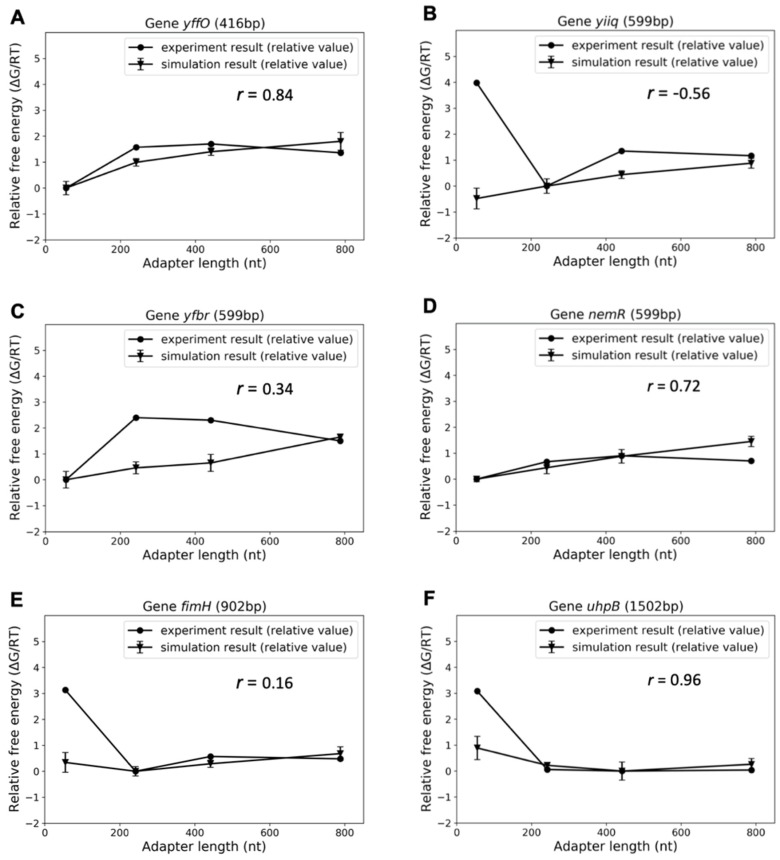
Comparison between free energy of probe–target interaction from experiments and simulation results. **A**–**F** represent six groups for data for six genes. The Pearson correlation coefficients are labeled as *r* value in the figures.

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
