# Peer review of "Computational Simulation of Adapter Length-Dependent LASSO Probe Capture Efficiency"

_biomolecules, 2019, doi:10.3390/biom9050199_

Round 1
Reviewer 1 Report
In the paper “Computational simulation of adapter length-dependent LASSO 2 probe capture efficiency” by Liu et al, the authors compare coarse grained simulation capture efficiencies with the free energy of target-probe interaction as estimated by RPKM measurements. The long-adapter ss oligonucleotide probes for multiplexed cloning of kilobase genome regions is a relatively new strategy for assembling larger segments of DNA than currently convenient by direct synthesis. The comparison with these experiments and simulation, while not straightforward, is novel and will be of interest to the readership.
1) The Geissler method is Virtual Move MC not Visual Move MC
2) While the RPKM results must involve free energy of association it is a bit less clear one can use the log ratios to directly produce a relative free energy. The authors were careful about the phasing but even more caution should be given to the reader.
3) The results in figure 4 are a bit disappointing whereas the enrichment data of figure 1 is clear.
4) While free energies were calculated from oxDNA simulations it seems that conformational entropy could be a strong effect in the overall process with respect to length.
Author Response
Dear reviewer,
Thanks for your time reviewing this and giving comments. We revised the paper according to your valuable suggestions and provide a point-by-point response to you. The response and the main changes in the revised paper are included in the attached PDF file. Please kindly find it in the attachment.
Best,
Jingqian Liu

Reviewer 2 Report
Authors provide an interesting manuscript entitled "Computational simulation of adapter length-dependent LASSO probe capture efficiency" addressing the relationship between the long-adapter length of LASSO and target capture enrichment. The authors suggest that the adapter length is a key feature that influences the efficiency of gene capture and cloning.
Main issue: the main conclusion is set based in the experimental analysis of only 3 different long adapters sequences. Independently of the results presented in the manuscript obtained by computational simulation it is advisable a more exhaustive study or control, not only in the number of different sizes analysed, as well as it association with the other constitutive proprieties associated to the adapter and target (i.e. % strong nucleotides; melting temperature ...).
It is also suggested to add an useful mathematical model for prediction of the size of adapter sequence according to the size of target of interest.
Minor Issues:
1- Taxonomic nomenclature and style along document Ex: E. Coli -> E. coli ;
2- Please revise order of figures as well as its quote along the documents;
3- Caption Figure 2 - Please provide a Comprehensive caption;
4- Please provide rational for choosing and test the mentioned: 242, 442 and 788nt long adapters sequences;
5- Please provide rational of the genes chosen to test RPKM;
6- Line 83 - Please disclosure or clarify "PCR primers TioINewRoche & SapINew (5-Phos)";
7- Line 90 - Please disclosure the number of replicates as well as the targets used for enrichment;
8- Line 101 - Please clarify sentence "The parameters ...>102.216nm, corresponding to Q values that range from 0 to 10", as well as the Q ex / Q Lig ratio as also shown in figure;
9- Line 281 - Please provide methods as well as data (supplemental material) regarding the determination of RPKM, shown here;
10 - Please revise supplemental file regarding stylus and leading autoexplanatory captions.
Author Response
Dear reviewer,
Thanks for your time reviewing this and giving valuable comments. We revised the paper according to your comments and provide a point-by-point response to you. The response and the main changes in the revised paper are included in the attached PDF file. Please kindly find it in the attachment.
Best,
Jingqian Liu

Round 2
Reviewer 2 Report
Minor issues:
Supplement material: Complete or reformat table "Primers used in the assembly of LASSOs"
Author Response
Dear reviewer,
Thanks for your quick review and the suggestion. We have revised the supplementary files and improved our manuscript. Please kindly find the specific response in the attached PDF file.
Jingqian Liu
